# Nutritional Support: The Use of Antioxidants in Inflammatory Bowel Disease

**DOI:** 10.3390/ijms25084390

**Published:** 2024-04-16

**Authors:** Sara Jarmakiewicz-Czaja, Katarzyna Ferenc, Aneta Sokal-Dembowska, Rafał Filip

**Affiliations:** 1Institute of Health Sciences, Medical College of Rzeszow University, 35-959 Rzeszow, Poland; sjczaja@ur.edu.pl (S.J.-C.); asokal@ur.edu.pl (A.S.-D.); 2Institute of Medicine, Medical College of Rzeszow University, 35-959 Rzeszow, Poland; kferenc@ur.edu.pl; 3Department of Gastroenterology with IBD Unit, Clinical Hospital No. 2, 35-301 Rzeszow, Poland

**Keywords:** antioxidants, Crohn’s disease, inflammatory bowel diseases, minerals, ulcerative colitis

## Abstract

The problem of treating inflammatory bowel disease continues to be a topic of great interest for researchers. Despite the complexity surrounding their treatment and strategies to prolong periods of remission, there is a promising exploration of various compounds that have potential in combating inflammation and alleviating symptoms. Selenium, calcium, magnesium, zinc, and iron are among these compounds, offering a glimpse of hope in the treatment of IBD. These essential minerals not only hold the promise of reducing inflammation in these diseases, but also show the potential to enhance immune function and possibly influence the balance of intestinal microflora. By potentially modulating the gut microbiota, they may help support overall immune health. Furthermore, these compounds could play a crucial role in mitigating inflammation and minimising complications in patients with IBD. Furthermore, the protective effect of these compounds against mucosal damage in IBD and the protective effect of calcium itself against osteoporosis in this group of patients are notable.

## 1. Introduction

Crohn’s disease (CD) and ulcerative colitis (UC) belong to the group of inflammatory bowel diseases (IBD) with a chronic course with periods of exacerbation and remission of the disease. In Western countries, the incidence is 12 to 26 per 100,000 people. Kaplan et al. distinguish four epidemiological stages of IBD: onset of IBD, acceleration of the disease, complex prevalence, and balance of prevalence [1]. Wang et al. in their study assessed the burden of IBD in 204 countries. They found that the incidence increased by more than 47% between 1990 and 2019. In addition, there were 68% more deaths from IBD in 2019 compared to 1990 [2].

Hospitalisation rates for patients with IBD vary in different countries according to the stage of the disease [3]. The aetiology of IBD is still under investigation. Researchers point, among others, to disorders of the immune system, genetic disorders, and intestinal dysbiosis with coexisting environmental factors [4,5,6]. In IBD, during inflammation, increased production of reactive species has been shown, and the body’s ability to cope with redox dyshomeostasis is impaired [7]. Polymorphisms of certain enzyme genes responsible for antioxidant processes, among others, can be linked to an increased risk of IBD, for example, promoter polymorphisms of SOD2 (superoxide dismutase-2) or Nuclear factor-erythroid 2-related factor 2 (Nrf2) promoter polymorphisms. Furthermore, intestinal epithelial cells and mucosal immune cells play an essential role in oxidative stress in IBD. Furthermore, tobacco smoke, psychological stress, antibiotic use, or certain food components, i.e., saturated fatty acids (SFA), refined sugar, and reduced antioxidant supply, can play a role in oxidative stress and thus exert an inflammatory effect on IBD [8,9]. During oxidative stress, the body generates not only reactive oxygen species (ROS), but also reactive sulphur species (RSS), reactive carbonyl species (RCS), and reactive nitrogen species (RNS). Reactions between these forms and downstream biological targets are defined as the reactive species interactome (RSI). Thus, RSI is related, among other things, to the nutritional status of the organism [10,11].

Many factors, including recurrent inflammation in inflammatory bowel disease, disrupt redox homeostasis. The body copes through the action of internal antioxidants such as superoxide dismutase (SOD), glutathione peroxidase (GPX), catalase, but also external antioxidants, including minerals [12].

The main objective of the review is to present the effects of selected minerals as potential antioxidants on IBD and to highlight the need to adjust the appropriate dose of supplementation.

The search for studies took place between July 2023 and February 2024 and included mainly full-length studies in English published in the PubMed database. The range of publications searched was 2000–2023 and used the MeSH terms “inflammatory bowel disease/dietary therapy”, “inflammatory bowel disease/immunology”, “inflammatory bowel disease/pathology”, “inflammatory bowel disease/prevention and control”, and “inflammatory bowel disease/therapy”.

## 2. Antioxidant Effect of Minerals

### 2.1. Selenium

Selenium exists in inorganic (selenate and selenite) and organic (selenomethionine and selenocysteine) forms [13]. The bioavailability of selenium largely depends on the chemical form. Organic forms are used for the biosynthesis of selenoproteins found in the body. Its bioavailability is also affected by the presence of food components such as proteins, fats, and heavy metals [14].

The recommended daily intake of selenium is 55 µg/db for adult women and men. Selenium requirements increase during pregnancy and lactation to 60 µg and 70 µg/day, respectively [15]. According to the Food and Nutrition Board (FNB), the upper tolerable intake of selenium is 400 µg/day (for pregnant women, it is 250 µg/day) [13,15]. However, there are data suggesting that adverse effects may occur with a selenium intake of 330 µg/day. It is believed that exceeding the upper tolerable intake level (UL) may occur as a result of oversupplementation or a high regular consumption of Brazil nuts [13].

The amount of selenium in food depends largely on the geographical area where the product is grown. Good sources of selenium are products such as Brazil nuts (originating from the eastern part of the Amazon, central Brazil, and the western part of the Amazon), fish (yellowfin tuna, halibut, and sardines), seafood, and crops [15,16]. Other sources include grain products, including bread, pasta, and rice, as well as meat, poultry, and eggs. Fortified products can be an equally good source of selenium in the diet [4].

Selenium as a dietary supplement is most often found in the form of selemethionine, a yeast grown in a medium with a high concentration of selenium, and also in the form of sodium selenate or sodium selenite [15].

Selenium is found in organs associated with the immune system, such as lymph nodes, bone marrow, spleen, liver, and thymus. At the cellular level, Se can affect various effector functions of leukocytes. Selenium acts through selenoproteins and some of them are involved in antioxidant defence (including glutathione peroxidase, thioredoxin reductase, and selenite synthase 2) and affect the immune system [17,18,19]. Analysis by Zhu et al. showed that many selenoproteins have a significant impact on the immune system and its specific functions [17].

Selenoproteins are important in innate and acquired immunity. They are responsible for cell activation and differentiation [17,20]. Table 1 shows the selenoproteins considered the most complete [17].

#### 2.1.1. The Effect of Selenium on the Microbiome

Selenium is known primarily for its effects on the course of diseases such as diabetes and cardiovascular and neurodegenerative disorders [23]. It is believed that selenium may also be a nutritional modulator of intestinal flora. However, there is still a lack of knowledge about the effect of Se on the intestinal microflora in humans, as most of the available data come from studies on animal models. Selenium may act together with other minerals as critical cofactors for bacterial enzymes responsible for antioxidant activity [14]. In a study by Lv et al. conducted on an animal model, the supply of probiotics enriched with selenium improved antioxidant capacity and reduced gastrointestinal symptoms [24]. Zhai et al. observed the effect of dietary Se supplementation on the intestinal barrier and the immune responses associated with the modulation of intestinal microflora. Inadequate selenium supplementation could lead to an altered gut microbiota composition that is more prone to developing colitis induced by dextran sulphate sodium and infections like *Salmonella typhimurium*. The authors of the study suggest that optimal or higher-than-normal selenium intake can help in shaping the gut microbiota to better defend against intestinal issues [25].

In a study involving overweight and obese women, the supply of Brazil nuts combined with a reducing diet influenced the growth of beneficial bacteria such as *Rumino-coccus*, *Roseburia*; NK4A214 and UCG-002 strains from the *Ruminococcaceae* family; as well as *Lachnospiraceae*, *Bacteroides*, and *Lachnoclostridium*. In addition, an improvement in the alleviation of intestinal permeability was observed [26].

Selenium, after being absorbed in the gastrointestinal tract, is transported to the liver where it is used to produce selenoproteins [27,28]. Takahaski et al. showed that selenocysteine and selenocyanate can be metabolised to selenomethionine (SeMet) by the intestinal microbiota, suggesting that seleno compounds can be metabolised to SeMet, which can be used by the host body [28]. Research results from recent years indicate that insufficient dietary selenium supply can lead to a competitive mechanism between the host and the microflora, so this should be taken into account when planning the daily requirement for this trace element [27,28]. Dietary selenium deficiency and consequently low levels of selenoproteins in the body affect various signalling pathways involved in inflammation, oxidative stress, and changes in the gut microbiota [29]. Furthermore, a bidirectional relationship has been observed between selenium levels in the body and the intestinal microflora, since it can also affect the selenium status in the body [30]. Interestingly, some species of intestinal microorganisms can improve the bioavailability of Se and protect against toxicity caused by high doses of Se supplements [14].

#### 2.1.2. Selenium and Inflammatory Bowel Disease

Patients with IBD show a decrease in SEPP1 (exhibits reductase and peroxidase activity) and SELENOP, especially in patients with Crohn’s disease [31]. In addition, SELENOP also correlates with the inflammatory response [14]. A study by Barrett et al. in a mouse model showed a protective effect of selenium against mucosal damage by reducing the expression of pro-inflammatory cytokines. The authors concluded that restoring normal Se levels may be helpful in the treatment of IBD, reducing inflammation and the risk of developing colon cancer [32]. Hiller et al. obtained different results. They showed that short-term selenite supplementation during acute colitis exacerbated the severity. Selenium supplementation is likely to have an anti-inflammatory effect under conditions of severe inflammation with significant selenium deficiency. As the authors point out, severe Se deficiency can affect selenoprotein expression in immune cells, thus exacerbating inflammation [33].

Inflammatory bowel diseases are characterised by significant intestinal dysbiosis and, as a consequence, an increase in pro-inflammatory bacteria, including *Escherichia* or *Fusobacterium*, and a decrease in species with anti-inflammatory effects, such as *Clostridium leptum*, *Faecalibacterium prausnitzii*, *Bifidobacterium*, or *Roseburia Clostridium coccoides* [14]. In addition, a reduced number of *Faecalibacterium prausnitzii* has been observed in the faeces of patients with Crohn’s disease (especially in patients with ileal involvement) [34]. Supplementation with selenium is likely to influence the restoration of the normal intestinal microbiota and the growth of *F. prausnitzii*, and thus alleviate the course of the disease [35]. *F. prausnitzii* acts on dendritic cells, which stimulate Treg cells to produce anti-inflammatory cytokines [36]. *F. prausnitzii* plays an important role in the supply of energy to colonocytes and its low levels negatively correlate with disease activity in patients with UC [34].

Patients with IBD (in adults and children) commonly have low serum selenium levels [37,38]. Furthermore, the intake of selenium with the diet in this group of patients is also significantly lower compared to healthy controls [39]. Selenium deficiency is believed to be associated with the severity of IBD [29]. According to Nettleford et al., Se supplementation can alleviate inflammation and increase pro-resolving pathways; therefore, it can be used in the treatment of IBD to alleviate symptoms [40]. A summary of the effect of selenium on the course of IBD is presented in Figure 1 [29,32,40].

### 2.2. Calcium

The World Health Organization (WHO) recommended daily calcium intake for adults is 1000 mg/d, that for postmenopausal women is 1300 mg/d, and during pregnancy the additional calcium supply is estimated at 420 mg/d. According to the Scientific Committee for Food, the upper daily tolerable intake for calcium is 2500 mg/d [41].

The main sources of calcium in food include dairy products, green vegetables such as broccoli, spinach, chard, as well as legumes, seeds, nuts, fish with small bones, and fortified foods [41,42].

Calcium absorption is also affected by nutritional and non-nutritional factors. These include nutrients such as iron, sugars (fructose), and organic compounds, among them oxalates, tannins, and phytates [43].

Animal model studies have shown that a diet containing calcium (39.6 g/kg dry weight; Ca^+^) can affect the peripheral and intestinal immune systems. Reduced Ca concentration can negatively affect acquired cellular immunity. Hens fed calcium-supplemented diets had higher levels of T cells and CD4^+^ lymphocytes [44]. Animal cell studies have shown an inhibitory effect of calcium citrate on NO production by suppressing cytokine expression cytokines, i.e., tumour necrosis factor alpha (TNF-α), interleukin-1 beta (IL-1β), and interleukin-6 (IL-6), and pro-inflammatory mediators nuclear factor kappa-light chain enhancer of activated B cells (NF-κB), inducible nitric oxide synthase (iNOS), and cyclooxygenase-2 (COX-2) in lipopolysaccharide (LPS)-stimulated RAW 264.7 macrophages [45]. In a study by Wang et al., the addition of calcium citrate did not affect interleukins and TNF-α [46]. On the other hand, the use of calcium lactate significantly reduced the levels of cyclooxygenase-1 (COX-1) and COX-2 in the liver [47]. In a study by Cha et al., the use of a higher dose (12 g/kg) of calcium citrate had a weight reduction effect with a similar energy intake of the diet. Furthermore, lower levels of white epididymal adipose tissue were observed, but plasma levels of triglycerides, insulin, and adiponectin were not significantly different between the groups. High calcium carbonate and calcium citrate improved metabolic disturbances (i.e., plasma leptin and glucose levels) that were induced by the high-fat diet. Furthermore, calcium citrate showed better efficacy, which may depend on the degree of bioavailability. Calcium is likely to affect fat oxidation and appetite control [48]. However, data on the role of Ca in the regulation of the immune system in humans are still lacking.

#### 2.2.1. The Effect of Calcium on the Microbiome

Calcium can affect the composition of the microbiota by influencing intestinal pH, acidifying the environment, and acting as a preservative and buffering agent [48]. A study by Cha et al. observed an increase in propionate-producing bacteria, which may contribute to improving intestinal barrier function, reducing oxidative stress and therefore inflammation [48]. In a study by Tong et al., the supply of sodium propionate affected the inhibition of the expression of interleukins IL-6, IL-1β, and TNF-α [49]. An increase in the concentration of lactic, succinic, acetic, propionic, and butyric acid after the use of dietary calcium phosphate in adult Wistar rats was also observed by Fuhren et al. [50]. In the same study by Wang et al., the use of calcium citrate and organic trace had an effect on the abundance and diversity of microorganisms in the faeces. In addition, the supply of calcium citrate increased the abundance of *Phascolarctobacterium* and *Lachnospiraceae* [46]. Ca-supplemented mice show higher levels of *Bifidobacterium* spp. and *Bacteroides* to *Prevotella* [51]. Hence, it is thought that calcium may have a prebiotic effect. It is likely that each of the *Bifidobacterium* strains acts specifically on different cytokines. *Bifidobacterium* has been found to affect the reduction of interleukin-12 (IL-12) and interleukin-8 (IL-8) [51].

A study involving women diagnosed with postmenopausal osteoporosis showed that the supply of the probiotic *Bifidobacterium* animalis subsp. Lactis Probio-M8 in combination with calcium and calcitriol may improve bone metabolism and anti-inflammatory indicators in this group of patients. Decreased levels of alkaline phosphatase and osteocalcin levels were noted [52].

In vivo and in vitro experiments are still necessary, especially with the participation of humans.

#### 2.2.2. Calcium and Inflammatory Bowel Disease

Since IBD patients are at risk of losing bone mass, developing osteoporosis, and hypocalcemia, calcium plays an important role in their diet. The optimal calcium supply in this group of patients is 1000–1500 mg/d. With a limited supply of dairy products, supplementation of this element in the diet should be considered [53,54]. The results of the study confirm that patients with IBD who declare lactose intolerance consume smaller amounts of calcium in their diet. In addition, women with UC consume significantly less calcium than the recommended daily intake [55]. A low supply of calcium with the diet or with inefficient absorption of calcium from the gastrointestinal tract results in its uptake from the skeleton [41]. As studies indicate, patients with CD consume insufficient amounts of calcium and their serum levels are inversely correlated with nutritional status and inflammation in the body, especially in the active phase of the disease, so it is believed to be one of the markers in the diagnosis of CD and the assessment of disease activity [56]. One strategy that may be helpful in the therapy of IBD may be the use of calcium pyruvate monohydrate (CPM), a stable pyruvate derivative. Pyruvate, as a key carbohydrate derivative, may act as an endogenous scavenger of reactive oxygen species and may exhibit anti-inflammatory and immunomodulatory effects. Therefore, CPM is believed to be part of an effective therapy for the treatment of IBD. It has also been shown that CPM can have systemic as well as local effects; as a result of its properties, it does not dissociate completely over a wide pH range, thus penetrating the intestinal mucosa more slowly. Studies in a mouse model have shown inhibitory effects of CPM on interleukin 17 (IL-17) and NF-κB expression (it is crucial in intestinal inflammation) [57]. Figure 2 shows the role of calcium on the intestinal microbiota discussed above and its potential impact on the course of IBS.

### 2.3. Zinc

Zinc (Zn) is one of the trace elements in the human body that is essential for many biological processes [58]. In the human body, it is present in an amount of 2–3 grammes. Approximately 57% is found in the skeletal muscles and 29% in the bones [59]. Oysters and red meat are significant dietary sources of zinc [60]. In the human diet, one of the main sources of zinc is grain products [61]. Other plant-based sources of zinc include nuts, seeds, and legumes [62]. Phytates present in grains and legumes can reduce its absorption, but soaking, fermentation, or acidification can increase bioavailability [63]. Additionally, sulphur-containing amino acids and organic acids in other foods increase zinc absorption [64]. The regulation of zinc homeostasis in the human body is controlled by intestinal absorption, involving transporters located on the apical and basolateral membranes of enterocytes [65]. The recommended daily allowance (RDA) for women is 8 mg per day, and for men, it is 11 mg [66]. These recommendations can vary according to factors such as country of origin, physical activity, health status, or physiological condition [67]. The recommended intake of zinc for adults is 8–11 mg. During pregnancy, zinc requirements increase to 11 mg per day, while during lactation it is 12 mg per day. The upper tolerable intake level (UL) for zinc in adults is 40 mg [68].

Due to the fact that there is no reserve in the body, a constant supply of this nutrient is necessary from food [69]. Zinc deficiency is estimated to occur in 17% to 26% of the population but is significantly more prevalent in African and Asian countries [58]. Zinc deficiency is observed in elderly people, people on a vegetarian diet, and patients with diseases such as IBD, liver cirrhosis, or kidney diseases [70,71,72,73]. In the literature, zinc deficiencies are divided according to the severity of the deficiency. Severe deficiency is most often encountered as a result of abnormal uptake in the intestine. This is most frequently encountered in patients with chronic diarrhoea, those taking parenteral nutrition, those using penicillamine, and alcohol abusers [60]. Zinc deficiency plays a significant role in impairing the immune system by reducing the number of T and B lymphocytes, resulting in increased susceptibility to infection [74].

The essence of the action of this element is the intracellular and extracellular regulatory properties of Zn^2+^, which interacts with many proteins [67]. Zinc has an important function as an enzyme cofactor, involved in the regulation of macronutrient metabolism, such as that of carbohydrates, proteins, and fats [75]. In addition, it is widely important for fibrinolysis, coagulation, and anticoagulation processes [76,77]. Furthermore, recent indications suggest that its role in the central nervous system is increasing, especially in basic physiological functions such as sleep and memory [78].

Zinc is not redox active, so its antioxidant role comes from indirect mechanisms of its action. It is a cofactor of the antioxidant enzyme superoxide dismutase but also increases glutathione (GSH) production [12,59]. As a potent endogenous antioxidant, SOD is responsible for converting the superoxide radical into hydrogen peroxide (H_2_O_2_) and molecular oxygen (O_2_) [79]. The excessive and uncontrolled formation of H_2_O_2_ can be toxic to cells [80]. Zinc is an antagonist of iron and copper. As an ion, it can displace these elements from specific binding sites, so it can mitigate the generation of (ROS) [81]. In vitro studies have shown that zinc reduces the activation of NF-kB as well as TNF-α and IL-1β [82]. Ranaldi et al. showed that the use of intracellular Zn is essential to maintain the integrity of the intestinal mucosa when exposed to TNF-α [83].

#### 2.3.1. Effect of Zinc on Gut Microbiota

Proper zinc homeostasis influences the maintenance of the normal structure and the role of the intestinal mucosal [84]. Disruption of this balance can contribute to intestinal dysbiosis, which in turn contributes to the development of IBD, irritable bowel syndrome, or colon cancer [85]. Furthermore, zinc is involved in the normal activity of alkaline phosphatase, which is involved in maintaining the integrity of the intestinal membrane [86]. In turn, Zackular et al. note that an excess of zinc can adversely affect the gut microbiota, potentially increasing the risk of *Clostridioides difficile* infection. These bacteria can exacerbate symptoms and increase mortality [87]. In addition, zinc can protect cells from tumour necrosis factor, which has a damaging effect. In addition, zinc can inhibit transcription factors that respond to oxidative stress caused by inflammation. This inflammation can promote disruption of intestinal epithelial integrity [88].

#### 2.3.2. Zinc and Inflammatory Bowel Disease

Intestinal permeability problems are one of the hallmarks of IBD patients. Sturniolo et al. showed that zinc supplementation can significantly reduce intestinal barrier permeability [89]. It turns out that zinc is crucial in maintaining the integrity of the intestinal epithelium, especially in the presence of damage that has been induced by cytokines [90]. Researchers showed that Zn supplementation in rats with induced colitis reduced diarrhoea and weight loss [91]. Patients with CD who are fed parenterally are found to be at very high risk of developing zinc deficiency, as a result of which they can develop visual disturbances and enteropathic dermatitis [92]. In turn, stunted growth, impaired vision and taste, and impaired immune function have been observed in children with CD along with zinc deficiency. Ishikara et al. report that zinc deficiencies in children with CD are significantly lower than in those with UC [93]. Siva et al., in a prospective study involving 773 patients with IBD, found that zinc deficiency in these patients was correlated with an increased risk of surgery, complications, and subsequent hospitalisations. When zinc levels were equalised, the results improved. The researchers emphasise the important role of monitoring the levels of this element, especially in patients with exacerbations and implementing appropriate supplementation at the time of deficiency [72]. Similarly, Moon et al. recommend screening for possible deficiencies in patients with active IBD [94]. Both zinc deficiency and zinc overload can contribute to the generation of oxidative stress [95]. Therefore, more research is necessary to fully exploit the antioxidant potential of zinc. Figure 3 shows the relationship between zinc and its effect on IBD.

### 2.4. Magnesium

Magnesium (Mg), as the second most abundant intracellular cation after potassium, is essential for the proper functioning of the body [96]. In addition, it is the fourth most important mineral in the human body [97]. Magnesium, which occurs as Mg^2+^, is found in the body in amounts of 22 to 26 g [98]. It is estimated that more than 99% of its amount is in the intracellular space, which is concentrated in bones (60–65%), soft tissues, and muscles (34–39%). The remaining 1% is in the extracellular space [99]. Depending on age, adults need 310–420 mg/day for magnesium [100,101]. This requirement may increase during pregnancy, in the elderly, during increased physical activity, and during the occurrence of infections or chronic diseases [102].

Foods rich in magnesium include nuts, green leafy vegetables, and whole-grain products [103]. Magnesium homeostasis in the human body is regulated by three mechanisms, including intestinal absorption, renal transport and excretion, and a mechanism that allows magnesium to be stored in bones [104]. In addition, this balance can be affected by caffeine and alcohol, a diet rich in calcium and sodium, and also by drugs such as proton pump inhibitors, diuretics, or antibiotics [105,106]. Because serum magnesium concentrations do not reflect actual magnesium saturation, most people with hypomagnesemia remain undiagnosed [107]. Despite a reference range of serum magnesium concentrations of 0.75–0.95 mmol/L, subclinical magnesium deficiency can occur [108]. Hypomagnesemia can also occur during the chronic use of proton pump inhibitors [109]. Medical conditions that may affect magnesium levels include alcoholism, unstable diabetes, malabsorption (e.g., CD, UC, short bowel syndrome, and celiac disease), endocrine diseases (e.g., hyperthyroidism and hyperparathyroidism), and kidney disease [110]. Trapani et al. showed significant magnesium deficiency in patients with IBD, and magnesium levels were inversely correlated with disease activity. Also, they showed that in mice, magnesium deficiency exacerbated colitis induced by sodium dextran sulphate (DSS) [111].

Magnesium has several functions in our body. One of them is being a cofactor for more than 600 enzymes and an activator for another 200 enzymes [112]. Furthermore, magnesium is a necessary cofactor for vitamin D, which in turn can increase the absorption of Mg from the intestine. Disruption of this homeostasis can potentially cause disorders such as bone deformities, metabolic syndrome, and cardiovascular disorders [113]. Magnesium is also crucial in nerve conduction, normal skeletal muscle, and cardiac function, but also in insulin and glucose metabolism [114]. Magnesium modulates lymphocyte proliferation and development, influencing acquired immunity [115]. Mg is essential for the proper functioning of all cells in the body because it is required for the synthesis of deoxyribonucleic acid (DNA) and ribonucleic acid (RNA).

In addition, magnesium maintains normal antioxidant levels in the cell [100]. Mg deficiency induces an inflammatory response, which is responsible for the release of pro-inflammatory cytokines, the activation of leukocytes and macrophages, and the production of acute phase proteins and ROS [116]. Liu et al. showed that magnesium supplementation can reverse this dysfunction by reducing mitochondrial ROS production [117]. Magnesium deficits may contribute to increased calcium (Ca) concentrations in mitochondria. This excess induces the production of pro-inflammatory cytokines, the activation of nitric oxide synthase (NOS), and the activation of the calcium-dependent calmodulin complex. This in turn induces ROS production [118].

#### 2.4.1. Effect of Magnesium on Gut Microbiota

Jørgensen et al. showed that a magnesium-deficient diet in rats alters their gut microbiota, resulting in anxiety-like behaviour [119]. On the contrary, García-Legorreta et al. proved that Mg supplementation above the recommended intake in mice without deficits can cause intestinal dysbiosis [120]. Xia et al. showed that the administration of magnesium isoglycyrrhizate (MgI) in mice contributed to the attenuation of methotrexate (MTX)-induced intestinal and liver damage through the regulation of intestinal barrier function. MgI upregulated *Lactobacillus* and *Muribaculaceae* bacteria [121]. Cao et al. showed in a study in piglets that potassium and magnesium sulphate supplementation can alleviate the incidence of diarrhoea but also modulate antioxidant defences, which appears to be due in part to regulation of the colonic microbiota [122]. Del Chierico et al. showed that magnesium can alleviate colitis in mice by increasing short-chain fatty acids (SCFA) and reducing the number of bacteria involved in inflammation [123]. Omori et al. in a study in mice showed that MgO along with inulin administration increased intestinal pH but also decreased levels of SCFA and lactic acid [124]. Intestinal bacteria such as *Lactobacillus* spp. may have influence in increasing magnesium bioavailability [125,126].

#### 2.4.2. Magnesium and Inflammatory Bowel Disease

Espen reports that patients with IBD are at risk for magnesium deficiency. The most common causes include gastrointestinal insufficiency, which is the result of diarrhea, improper diet, reduced oral intake, and malabsorption [127]. Geerling et al. showed the prevalence of lower magnesium levels in UC patients, both in exacerbation and remission, compared to controls [128]. Gilca-Blanariu et al. evaluated the magnesium status in 37 IBD patients (12 CD, 25 UC) by measuring hair concentrations using scanning electron microscopy (SEM) and X-ray spectroscopy. Finally, they showed statistically significantly reduced Mg concentrations in IBD patients compared to healthy controls. Comparing the CD and UC group, significantly lower concentrations are observed in the former group. Moreover, the authors believe that the role of Mg in cell signalling or DNA repair systems may be related to the occurrence of fatigue in patients [129]. In turn, Soare et al. showed that low Mg levels in IBD patients contribute to low bone density, especially in women [130]. Given that Mg deficiency results in reduced osteoblast activity, patients may develop bone metabolism disorders [131]. However, the role of deficiencies, especially of Mg, is insufficiently studied. Future studies will be able to confirm or deny its role in intestinal diseases.

### 2.5. Iron

Iron is an element that is absorbed from the diet by enterocytes in the duodenum. In the first stage, iron reductase changes the state of oxidation of the element from Fe^3+^ to Fe^2+^ and then it is transported across the cell membrane by a transporter. In the next stage, it changes the oxidation state again to Fe^3+^ and combines with apoferritin. Iron ions are stored in the liver, spleen, and bone marrow [132,133]. In addition, iron exists in two forms: heme and non-heme. Heme iron is absorbed by the body at a rate of 20–30%, while non-heme iron is absorbed at a rate of less than 10% [134,135].

The best dietary sources of heme iron are meat, fish, and seafood, while that for non-heme iron includes cereals, legumes, and dark green vegetables [136].

In pregnancy, the iron requirement increases to 27 mg per day, while during lactation, it is 9–10 mg per day [136]. The upper tolerable iron intake level (UL), depending on age, is 40–45 mg per day [136].

Certain groups of people are at risk for iron deficiency, i.e., pregnant women, cancer patients, patients with gastrointestinal diseases such as IBD, and people with heart failure [136]. However, oral iron supplementation has an effect on the intestinal microbiota. Bielik et al. indicate that there is a decrease in lactic acid bacteria during this supplementation. Furthermore, in their review, they present that *Lactobacillus plantarum* 299v may increase absorption of non-heme iron [137]. Husmann et al. also find a potentially beneficial effect of prebiotics on iron absorption, but this absorption depends on several additional factors such as the dose, type, or timing of prebiotic intake [138]. There are approximately 3–5 grammes of iron in the human body, of which almost 2/3 of the total pool is in hemoglobin and 10% in myoglobin. Approximately 1–2 mg of the element is lost daily as a result of epithelial exfoliation, among other things. Therefore, iron is absorbed in the amount of 1–2 mg, while the requirement for an adult is much higher, due to the fact that, for example, for the synthesis of hemoglobin alone, 20–25 mg of the element is required [139].

Programmed cell death is also essential for proper homeostasis of the body. A variant of programmed cell death, defined in 2012, is ferroptosis. Iron plays an important role in this case. Some of the enzymes involved in catalysis require the presence of this element. In addition, it is involved in the production of lipid ROS [140,141].

When it comes to the antioxidant effect of iron, the research is contradictory. On the one hand, it is an essential element for life, while on the other hand, its metabolism is strongly linked to oxygen, which is why labile iron, which has been linked to the phenomenon of ferroptosis, is drawing increasing attention from researchers. Maintaining iron balance in the body is crucial, as both a deficiency of the element and its excess can contribute to disruption of redox homeostasis [142]. Iron is classified as an antioxidant [143]. Due to the chemical properties of the element, perishable iron can be involved in the formation of reactive free radicals and undergoes chelation [144]. Unregulated iron can participate in the Fenton reaction, generating the formation of free radicals [145]. One of the factors involved in controlling the breakdown or synthesis of ferritin is nuclear factor 2-related factor 2 (NRF2). Furthermore, NRF2 maintains iron homeostasis through vesicle-associated membrane protein 8 (VAMP8), nuclear receptor coactivator 4 (NCOA4), and E3 ubiquitin protein ligase 2 (HERC2) [146].

#### 2.5.1. Effect of Iron on Gut Microbiota

The effect of iron on the gut microbiota is significant. Xiao et al. in their review indicate that the metabolism of the element is closely related to the gut microbiota and it may be a bidirectional relationship [147]. Because iron deficiencies often occur in IBD patients, supplementation is used. Both iron oral supplementation and intravenous iron supplementation can affect changes in gut microbial composition and diversity [148]. After oral supplementation, lower abundance of *Ruminococcus bromii*, *Faecalibacterium prausnitzii* and higher abundance of *Bifidobacterium* is observed in IBD patients [149]. In addition, the number of bacteria such as *Proteobacteria* increases, while the number of *Lachnospiraceae* decreases, with which iron may exacerbate dysbiosis in IBD [150]. Due to the irritation and affected permeability of the intestinal barrier fromexcessive iron, epithelial cells secrete increased amounts of secretory leukocyte protease inhibitor (SLPI), which, in large amounts, can be a pro-cancer factor [151]. In addition, some microbial communities also show an association with parameters associated with ferroptosis [152].

#### 2.5.2. Iron and Inflammatory Bowel Disease

Iron deficiency is often observed in IBD. Causes of anemia can range from the appearance of bleeding, through altered absorption of the element, to the use of a deficient diet by patients [153,154]. Iron is essential for proper body function, but the level of the element should be within normal limits in the body. Both its deficiency and its excess can cause redox dyshomeostasis [155]. Lee et al. in their study analysed the effect of iron levelling on the gut microbiome. They showed that bacterial diversity changes significantly after iron replacement therapy (IRT) and that CD patients were more susceptible to experiencing such changes [148]. Increasingly, researchers are turning their attention to ferroptosis inhibitors as agents that can potentially reduce inflammation, which could be beneficial for the therapeutic process [156,157,158]. This is why maintaining adequate iron levels in the body is so important. Given the prevalence of anemia, the European Crohn’s and Colitis Organisation (ECCO) has developed standards in which patients with IBD in remission have ferritin levels of <30 µg/L and up to 100 µg/L in exacerbation may indicate micronutrient deficiency and treatment should be instituted [159]. Both low serum iron concentrations and high doses may exhibit toxic effects on the epithelium due to increased inflammatory reactions [160]. Supplementation with excessive amounts of iron may also affect the epigenetic modifications with which it can modulate the condition of IBD patients [161]. Iron compensation in IBD patients can help protect against ROS damage and repair the intestinal barrier, but it is important to make sure supplementation is as recommended [162].

## 3. Limitations

A limitation of the review may be that there are too few human studies on the effects of selected minerals on the exacerbation and remission in patients with IBD. In addition, studies very often lack homogeneous groups of patients in terms of gender and medications used. There is also often no indication of the effects of excess antioxidants on the intestinal barrier, gut microbiota, and exacerbations in IBD patients, which could set future research directions.

## 4. Conclusions

A disturbed redox balance can lead to prolonged inflammation in the body of patients with IBD. In addition to internal antioxidants, it is extremely important to pay attention to external antioxidants. Adequate levels of components such as selenium, zinc, iron, and magnesium can support the primary therapy of patients with IBD. Deficiency and excess can further aggravate redox dyshomeostasis, so it is worth monitoring patients’ levels of these components and implementing supplementation if necessary. In addition to their primary antioxidant effect, they may also have an effect on the gut microbiota, both on the number and type of microorganisms. By understanding and addressing the role of minerals as antioxidants, healthcare providers can potentially improve quality of life and outcomes for people living with inflammatory bowel disease.

## Figures and Tables

**Figure 1 ijms-25-04390-f001:**
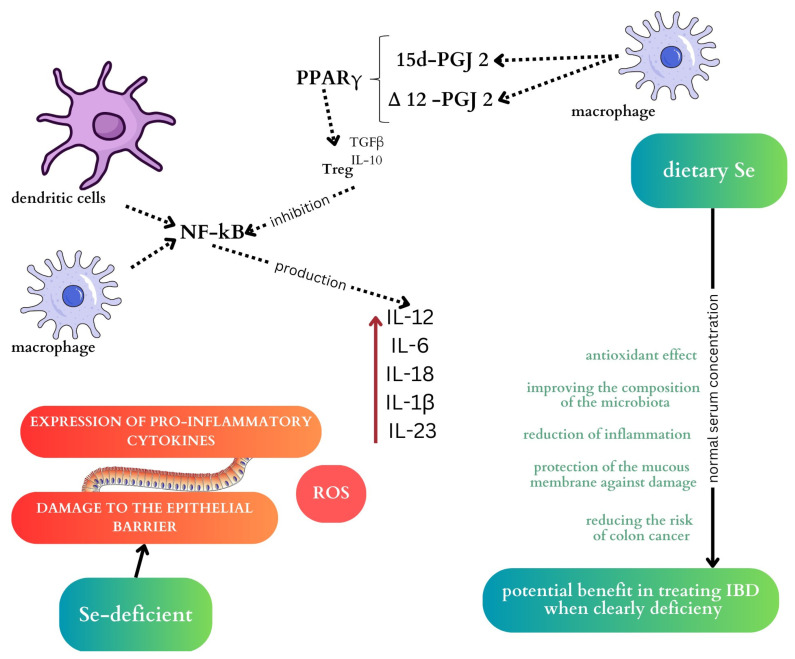
Diagram summarising the effects of selenium on the intestinal microbiota and IBD. Selenium plays several crucial roles in the body. It acts as an antioxidant, enhances the gut microbiota, and protects the intestinal barrier, potentially reducing inflammation. Selenium deficiency can disrupt nuclear factor κB signalling, triggering inflammation exacerbated by reactive oxygen species which harm tissue integrity by affecting the epithelial barrier. When the epithelial barrier is compromised, immune responses are activated. Macrophages and dendritic cells detect bacterial presence, initiating immune reactions. This cascade involves increased nuclear factor-kappa-B levels, prompting the release of pro-inflammatory cytokines. In the presence of selenium, macrophages generate 15d-prostaglandin J 2 (15d-PGJ 2) and ∆12-prostaglandin J 2 (∆12-PGJ 2). The nuclear hormone receptor, peroxisome proliferator-activated receptor gamma (PPARγ), is upregulated and moderates NF-κB, suppressing proinflammatory cytokine production. PPARγ promotes Tregs differentiation, curtailing T helper cell activity.

**Figure 2 ijms-25-04390-f002:**
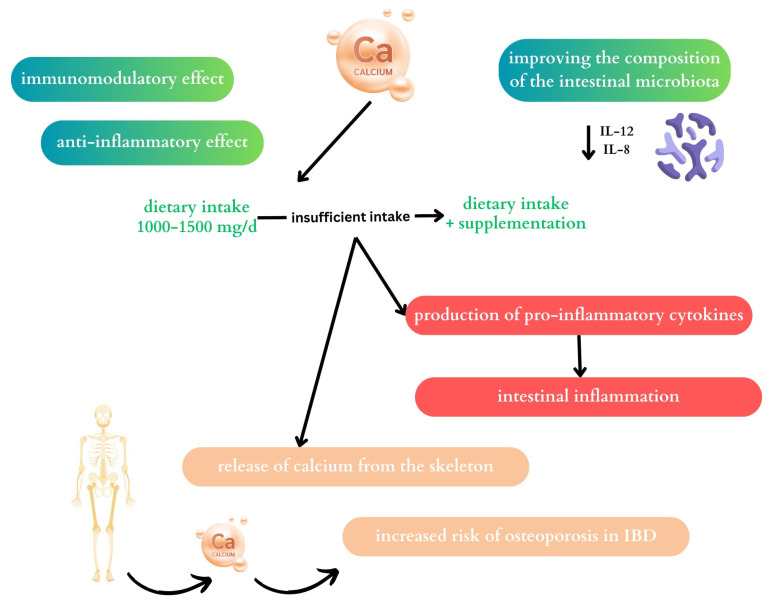
This diagram shows the potential impact of calcium in the treatment of patients with IBD. If there is an insufficient supply of calcium in the diet, this element may be released from the skeleton. Patients with IBD are at high risk of developing osteoporosis; therefore, if the dietary supply of calcium is insufficient, calcium supplementation should be considered. Calcium has a potential immunomodulatory and anti-inflammatory effect in patients with IBD. Additionally, it may improve the composition of the intestinal microbiota. Compensatory bacteria can reduce the level of pro-inflammatory cytokines in the body.

**Figure 3 ijms-25-04390-f003:**
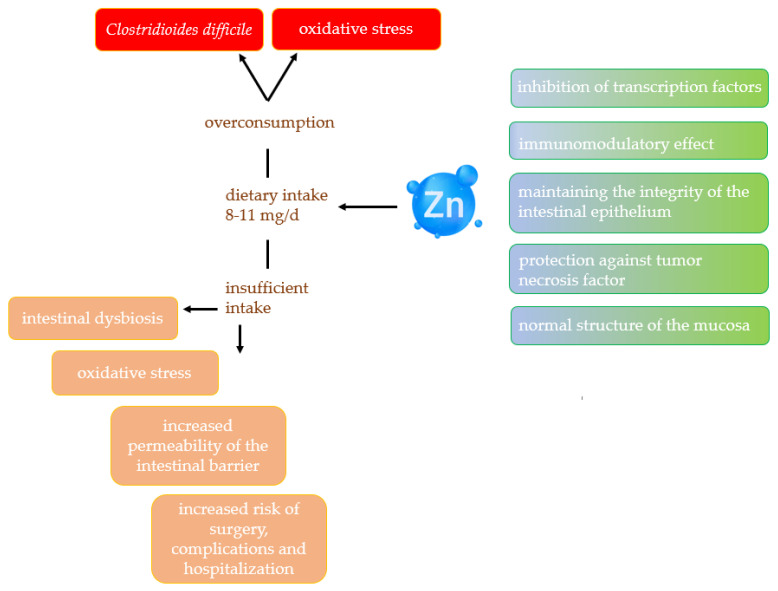
The relationship between zinc, microbiota, and IBD.

**Table 1 ijms-25-04390-t001:** Immunomodulatory functions of selenoproteins.

Selenoptrotein	Functions
GPXs	Promote hydroperoxide metabolism and reduce damage to the body [17].
GPX1	Key regulatory role in the cell apoptosis signalling pathway (mainly, in which damaged mitochondria produce ROS above a physiological level; involved in the activation and differentiation of T cells [17,21].
TXNRD	Acts as a regulatory factor and regulated target in macrophage gene expression [17].
DIO	Deficiency interferes with the conversion of T4 to T3 and may affect systemic selenium levels [17].
MSRB1	Responsible for DC-mediated T cell activation and Th1 differentiation; acts as an oxidoreductase [17].
SPS2	A catalyst for selenophosphate production, which is used to synthesise selenoproteins [20]
SELENOK	Promotes Ca^2+^ flow in immune cell activation; interacts with DHHC6; the SELENOK promotes palmioylation of proteins catalysed by DHHC6 [17,22]

GPX—glutathione peroxidase; GPX1—cytosolic glutathione peroxidase; TXNRD—thioredoxin reductase 1; DIO—deiodinase; MSRB1—selenoprotein R; SPS2—selenophosphate synthetase; SELENOK—selenoprotein K; T3—triiodothyronine; T4—thyrosine; DC—dendritic cells; ROS—reactive oxygen species; DHHC6 (letters represent amino acids aspartic acid, histidine, histidine, and cysteine in the catalytic domain).

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
