# Peer review of "Nutritional Support: The Use of Antioxidants in Inflammatory Bowel Disease"

_ijms, 2024, doi:10.3390/ijms25084390_

Round 1

Reviewer 1 Report

Comments and Suggestions for Authors

Reading the Abstract of the manuscript showed a very promising paper, with so many good effects of the included minerals on reducing symptoms, mucosal inflammation, mucosal damage, complications etc, therefore a real benefit in the therapeutic armamentarium for IBD patients. However, the whole manuscript appears rather disappointing. There are too many data about nutritional facts of these minerals (not the real topic), while effects on IBD are very scarce. I have listed my comments/suggestions for consideration, below:

1.       Introduction:

a.       Please mention also some sentences about IBD prevalence, which is markedly increasing, with incidence decreasing in some regions. The prevalence is the one that matters in this review, as it tackles about therapeutic approaches in established IBD and not about prevention.

b.       Please add reverence “Wang R, Li Z, Liu S, Zhang D. Global, regional and national burden of inflammatory bowel disease in 204 countries and territories from 1990 to 2019: a systematic analysis based on the Global Burden of Disease Study 2019. BMJ Open. 2023 Mar 28;13(3):e065186”

c.        Please, by the end of Introduction, clearly define the aim of your review. Since this is a review, please also mention the databases you searched, periods and respective MeSH.

2.       Antioxidant effect of minerals

a.       Generally, I found too many generalities about each mineral. I suggest the Authors to focus more on IBD, as this is the real topic of the review.

b.       I suggest removing the Tables with “Recommended Dietary Allowances (RDA) for xx”, for each mineral, as they can be found at references [14, 38 etc]. This review is about their effects in IBD and not a nutrition paper.

c.        Selenium

c1. I suggest removing the detailed description of selenoproteins mentioned in the paper by Zhu et al, and keeping just the ones that are important and described in Table 2. Everyone with special interest in the description of selenoproteins can look up in detail reference [16].

c2. Selenium and IBD: Please summarize and incorporate all the data in a figure and show the clear effects in IBD (inflammation, damage etc).

d.       Calcium

d1. Please remove all the sentences that do not clearly refer to the topic.

d2. Calcium and IBD – Please summarize and incorporate all the data in a figure and show the clear effects in IBD. Some sentences that are written in the general aspects about Calcium could be used here.

e.       Zinc

e1. Reference [62] refers to Polish population. Please insert a reference valid for general population. Could still be from https://ods.od.nih.gov/factsheets: https://ods.od.nih.gov/factsheets/Zinc-HealthProfessional/. Then, remove please Table 4, as I mentioned before.

e2. Also, too many sentences. You cannot see the forest for the trees.

e3. Zinc and IBD: Please elaborate more on the topic and then summarize in a figure.

f.         Magnesium

f1. Reference [62] refers to Polish population. Please insert a reference valid for general population. Could still be from https://ods.od.nih.gov/factsheets: https://ods.od.nih.gov/factsheets/Magnesium-HealthProfessional/. Then, remove please Table 5, as I mentioned before.

f2. Also, too many sentences that are not really needed. Please stick to your aim.

f3. Magnesium and IBD – very little useful info. Not truly useful and summarized.

f4. Please remove the study about chronic constipation, as this is not IBD; same for functional disorders involving gut-brain axis.

g.        Iron

g1. Too many nutritional info. Please remove Table 6, this is not a paper about healthy nutrition/special diets.

g2. Reference [62] refers to Polish population. Please insert a reference valid for general population. Could still be from https://ods.od.nih.gov/factsheets: https://ods.od.nih.gov/factsheets/Iron-HealthProfessional/. Then, remove please Table 7, as I mentioned before.

g3. Iron and IBD: This is almost all about iron deficiency. Not really about the topic.

3.       Conclusions

a.       Please correct the misspelling and insert 3, not 7.

b.       This is too general, while proper effects that have been shown in the Abstract are lacking. Please revise the entire manuscript and provide a robust, clear conclusion with proper definite effects. Otherwise, the aim of the review was not fulfilled.

4.       References are generous, but they refer more at nutritional facts, then at the proper topic. Please revise and update.

5.       I read the iThenticate report in detail and it looks fine.

Comments on the Quality of English Language

 English language: quality is generally good, just some typos, misspelled words, non-agreement verb-noun.

Author Response

Dear Reviewer,

Responses to the following comments are attached.

Kind regards

Authors

Reviewer 2 Report

Comments and Suggestions for Authors

The manuscript by Sara Jarmakiewicz-Czaja et al. summarized recent study about the use of antioxidants in the treatment of inflammatory bowel diseases. The paper is informative and interesting. I have the following questions and comments:

1, all the table should be in a three-line format. The authors should double check. 

2, I suggest the authors to add a new figure to summarize the mechanisms underlying the beneficial effects of antioxidants in the treatment of inflammatory bowel diseases. 

3, current research limitations should be discussed and future directions should be provided. 

4, gut microbiota plays a pivotal role in the development of IBD. The effects of selenium, calcium, magnesium, zinc, and iron on the gut microbiota should be discussed separately. Human studies and animal studies should be included. 

Author Response

(The authors gave the same response as above.)

Round 2

Reviewer 1 Report

Comments and Suggestions for Authors

I congratulate the Authors for their markedly improved manuscript. What a big change! The text appears clearly organized, containing pertaining scientific information and the addition of the figures is remarkable. Now, everything reads very easily. Only one minor comment: I was asking the Authors: “Since this is a review, please also mention the databases you searched, periods and respective MeSH”. Database was inserted, but the start period of including studies – not. The authors did not include only studies between July 2023 and February 2024. Then, please indicate what was the first date of the papers you included. How long did you go back in time for the search? Regarding MeSH, they were not written. Please add. 

Comments on the Quality of English Language

Looks fine to me, just some minor typos.

Author Response

Dear Reviewer,

Thank you very much for your comment, the changes have been added to the manuscript.

Kind regards
Authors

Reviewer 2 Report

Comments and Suggestions for Authors

The authors have revised the manuscript accordingly. It can be considered for publication. 

Author Response

Dear Reviewer,

Thank you very much for your comment.

Kind regards
Authors